# Quantitative real-time imaging of glutathione

Xiqian Jiang[1,*], Jianwei Chen[1,*], Aleksandar Bajić[2,3], Chengwei Zhang[1], Xianzhou Song[1], Shaina L. Carroll[1,4], Zhao-Lin Cai[5,6], Meiling Tang[1], Mingshan Xue[5,6,7], Ninghui Cheng[2,8], Christian P. Schaaf[3,7], Feng Li[9,10], Kevin R. MacKenzie[10,11], Allan Chris M. Ferreon[1], Fan Xia[7], Meng C. Wang[7,12], Mirjana Maletić-Savatić[2,3,10] & Jin Wang[1,9,10]

Glutathione plays many important roles in biological processes; however, the dynamic changes of glutathione concentrations in living cells remain largely unknown. Here, we report a reversible reaction-based fluorescent probe—designated as RealThiol (RT)—that can quantitatively monitor the real-time glutathione dynamics in living cells. Using RT, we observe enhanced antioxidant capability of activated neurons and dynamic glutathione changes during ferroptosis. RT is thus a versatile tool that can be used for both confocal microscopy and flow cytometry based high-throughput quantification of glutathione levels in single cells. We envision that this new glutathione probe will enable opportunities to study glutathione dynamics and transportation and expand our understanding of the physiological and pathological roles of glutathione in living cells.

[1] Department of Pharmacology and Chemical Biology, Baylor College of Medicine, Houston, Texas 77030, USA. [2] Department of Pediatrics, Baylor College of Medicine, Houston, Texas 77030, USA. [3] Jan and Dan Duncan Neurological Research Institute at Texas Children's Hospital, Houston, Texas 77030, USA. [4] Department of Chemistry, Rice University, Houston, Texas 77030, USA. [5] Department of Neuroscience, Baylor College of Medicine, Houston, Texas 77030, USA. [6] The Cain Foundation Laboratories, Jan and Dan Duncan Neurological Research Institute at Texas Children's Hospital, Houston, Texas 77030, USA. [7] Department of Molecular and Human Genetics, Baylor College of Medicine, Houston, Texas 77030, USA. [8] USDA/ARS Children Nutrition Research Center, Baylor College of Medicine, Houston, Texas 77030, USA. [9] Department of Molecular and Cellular Biology, Baylor College of Medicine, Houston, Texas 77030, USA. [10] Center for Drug Discovery, Baylor College of Medicine, Houston, Texas 77030, USA. [11] Department of Pathology, Baylor College of Medicine, Houston, Texas 77030, USA. [12] Huffington Center on Aging, Baylor College of Medicine, Houston, Texas 77030, USA. * These authors contributed equally to this work. Correspondence and requests for materials should be addressed to J.W. (email: wangj@bcm.edu).

Glutathione (GSH) is the most abundant non-protein thiol in eukaryotic cells. Together with its oxidized partner (GSSG), GSH maintains cellular redox homeostasis[1], regulates protein functions through S-glutathionylation[2], contributes to iron-sulfur cluster maturation[3], and acts as a signalling molecule to directly activate gene expression[4,5]. These important functions are dynamically regulated by the intracellular concentration and distribution of GSH[6]. Currently, the concentration of intracellular GSH is derived from either cell lysates or GSH-S-transferase (GST) dependent probes[7,8]. These approaches, however, cannot provide information about the real-time dynamics of GSH concentration changes[9]. Redox-sensitive fluorescent proteins (roFPs) have been widely used to study GSH-related redox biology[10,11]. However, roFPs only measure the changes in redox potential ($E_{GSH}$) in cells, which can be due to shifts in the [GSH]:[GSSG] ratio, changes in total GSH concentration, or a combination of both scenarios. Many intracellular biochemical reactions, especially enzymes that use GSH as their substrate, rely on the local GSH concentration instead of $E_{GSH}$ (ref. 12). Besides, roFPs under physiological conditions are only sensitive to oxidative stress but not to any 'reductive stress' because they completely transition to a fully reduced state once in the cytosol. Herein, we developed a fluorescent probe—designated as RealThiol (RT)—that can quantitatively monitor the real-time GSH dynamics in living cells.

Small-molecule fluorescent probes have gained increasing attention since the emergence of calcium and zinc probes[13,14]. We reported the first reversible reaction-based small-molecule fluorescent GSH probe (ThiolQuant Green) that can perform single-point quantification of GSH levels in living cells[15]. The major obstacles to developing probes that can monitor GSH dynamics include the reaction reversibility and kinetics of the sensing reaction, as well as the high abundance of GSH (1–10 mM) inside cells. We thus designed RT (Fig. 1a) taking into account of all these challenges[16]. The reaction between RT and GSH is based on a Michael addition reaction that is inherently reversible, has an appropriate dissociation equilibrium constant $K_d$ in the mM range and rapid reaction kinetics, and provides ratiometric readouts, thus allowing GSH quantification independent of the probe concentration. In the RT structure, the cyano group at the α position of the Michael acceptor enables fast reaction kinetics, the four-membered azetidine ring improves quantum yield and photostability[17], and the two carboxylic acid groups ensure aqueous solubility and reduce probe binding to hydrophobic cellular structures. To enhance the cell permeability of RT, we converted the carboxylic acid groups to acetoxymethyl (AM) esters, which are readily hydrolysed by esterases to regenerate RT inside cells. Using RT, we were able to monitor the dynamic changes of GSH in living cells, which subsequently led to the observation of enhanced antioxidant capability of activated neurons and time-dependent changes of GSH during the ferroptosis process.

## Results

**Spectroscopic and physical characterizations of RealThiol.** RT shows ratiometric fluorescence responses with a wide dynamic range when reacting with GSH. RT and its GSH adduct (RT-GSH) shows fluorescence maxima at 487 and 562 nm with excitation wavelengths at 405 and 488 nm, respectively (Fig. 1b). Plotting the fluorescence intensity ratios with excitation wavelengths at 405 and 488 nm ($F_{405}/F_{488}$) as a function of GSH concentrations confers a superb linear relationship ($R^2 = 0.998$) covering the physiological GSH concentration range 1–10 mM (Fig. 1c, Supplementary Figs 1 and 2). The $K_d$ for the reaction between RT and GSH is 3.7 mM. Comparing to ThiolQuant Green, RT and RT-GSH has much improved quantum yields and photostability (Table 1 and Supplementary Fig. 3). Although

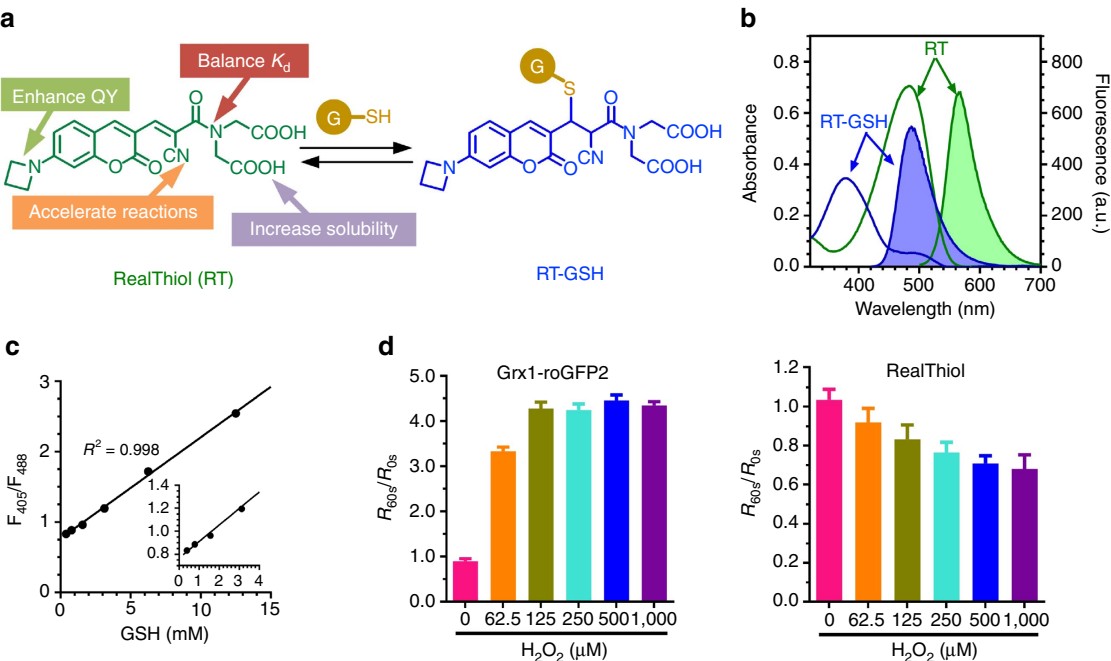

**Figure 1 | Characterization of reversible reaction-based glutathione probe RT.** (**a**) The reversible Michael addition reaction between RT and GSH. The function of each moiety in RT is highlighted (QY: quantum yield). (**b**) Ultraviolet–visible (unshaded) and fluorescence (shaded) spectra of RT (green) and RT-GSH (blue), the GSH adduct. (**c**) Linear relationship between $F_{405}/F_{488}$ and GSH concentrations. $F_{405}$ and $F_{488}$ are the fluorescence intensities for RT-GSH and RT, respectively. (**d**) Dynamic ranges of Grx1-roGFP2 and RT probe. Fluorescence ratio changes on treatment with different concentrations of $H_2O_2$ in HeLa cells is measured, $R_{60s}/R_{0s}$ is calculated as the ratio $R$ value 60 s after $H_2O_2$ treatment divided by the $R$ value before treatment. Each point is the mean value of >8 cells analysed from 2 independent experiments. Error bars represent s.e.m.

**Table 1 | Quantum yields of GSH probes.**

| Species | Quantum yield (%) | Solvent |
| --- | --- | --- |
| Rhodamine 123 (reference) | 97.0 | PBS |
| RT | 2.9 | 1% DMSO in PBS |
| RT-GSH | 86.0 | PBS |
| TQG | 0.94 | 1% DMSO in PBS |
| TQG-GSH | 0.59 | PBS |
| TQG-CN | 5.0 | 10% DMSO in PBS |
| TQG-CN-GSH | 12.5 | PBS |
| RT-NH$_2$ | 17.7 | 10% DMSO in PBS |
| RT-NH$_2$-GSH | >97.0 | PBS |

Quantum yields of GSH probes. RT has a high quantum yield and high photostability. Coumarin-based fluorescent probes usually suffer from low quantum yields (QYs). Substituting the $N,N$-diethylamino group with an azetidine can substantially improve the QY of RT to 86.0% ($\lambda_{ex} = 406$ nm, RT-GSH) and 2.9% ($\lambda_{ex} = 479$ nm, RT) compared to 0.59% ($\lambda_{ex} = 406$ nm, TQG-GSH) and 0.94% ($\lambda_{ex} = 479$ nm, TQG) for TQG. The data for TQG and TQG-GSH were obtained from ref. 15. TQG-CN and RT-NH$_2$ have very low aqueous solubility, therefore 10% DMSO is required for quantum yield measurements. Because quantum yields are highly dependent on solvents, the quantum yields between TQG-CN and TQG are not comparable. The quantum yield of RT-NH$_2$-GSH is higher than that of the reference Rhodamine 123.

roFPs are highly sensitive to oxidative stress, their responses can be easily saturated (Fig. 1d). In contrast, the wide dynamic range of RT allows for monitoring of the GSH level changes within 1–10 mM in both directions in living cells (Fig. 1d).

**Selectivity of RealThiol under physiological conditions**. RT preferentially reacts with GSH under physiological conditions. Due to both the high abundance of GSH and the millimolar $K_d$ value, we observed $F_{405}/F_{488}$ changes only when RT reacted with GSH but not other thiols, reactive oxygen or nitrogen species (ROS/RNS) at their corresponding physiological concentrations (Fig. 2a). The ratiometric readout of RT only responds to GSH (10 mM), but not to glycine (10 mM), cysteine (100 μM) or a series of ROS/RNS (100 μM). It should be noted that cysteine and GSH have similar chemical reactivities towards RT (Fig. 2c). Because intracellular cysteine level is usually below 100 μM under the physiological conditions[18,19], the interference of RT signals from cysteine is minimal. To assess whether RT probe binds to any proteins or reacts with free cysteine residues in some proteins that may interfere with the final ratio readouts, we incubated RT with cell lysate, which was dialyzed using a 3 K cut-off membrane to remove all small molecules including GSH, and observed no significant changes in fluorescent signal ratios even at the highest lysate concentration shown (protein concentration is 36 mg ml$^{-1}$ based on a BCA assay), suggesting that there is no non-specific interactions or reactions between cellular proteins and RT which could significantly change the ratio readouts (Fig. 2a). Furthermore, $F_{405}/F_{488}$ is insensitive to other environmental factors, including potential non-specific interactions or reactions with intracellular proteins (Fig. 2a), pH and viscosity (Fig. 2b). In addition, to determine the percentage distribution of reacted RT between GSH and protein thiols, we treated HeLa cells with RT and lysed the cells using trichloroacetic acid to separate proteins from small molecules (Fig. 2d). Gel permeation chromatography (GPC) analysis showed that 10% of RT reacts with thiolated proteins while 90% of RT reacts with GSH (Fig. 2e and Supplementary Table 1). Therefore, we conclude that RT is able to reliably measure the intracellular GSH levels.

**Reaction kinetics of RealThiol**. Unlike the currently available GSH probes[9,15], RT features fast kinetics in both forward and reverse directions, enabling real-time monitoring of GSH dynamics with a minute-level time resolution. To measure the forward reaction rates, we mixed RT (10 μM) with equal volumes of different concentrations of GSH solutions (5, 10 and 20 mM) at

pH 7.4 and monitored the formation of RT-GSH using a fluorimeter ($\lambda_{ex} = 405$ nm; Fig. 3a). The second-order reaction rate constant between RT and GSH is 7.5 M$^{-1}$ s$^{-1}$, while the same rate constant for our previously reported ThiolQuant Green probe and GSH is 0.15 M$^{-1}$ s$^{-1}$, which is 50 times slower than RT (Table 2). To measure the reverse reaction rates, we mixed a pre-equilibrated mixture of RT (20 μM) and GSH (10 or 5 mM) with an equal volume of PBS and followed the disappearance of RT-GSH fluorescence (Fig. 3a). The first-order dissociation reaction rate constant for RT-GSH is $20.3 \times 10^{-3}$ s$^{-1}$, which is more than 500 times faster than that for ThiolQuant Green-GSH ($35.7 \times 10^{-6}$ s$^{-1}$). In addition, it should be noted that GST has little catalytic effect on the reaction rate between RT and GSH (Fig. 3b).

**Applications of RealThiol in living cells**. RT quantitatively and reversibly responds to the fluctuations of intracellular GSH levels. For cells treated with the AM form of RT, real-time ratiometric images of the rapid changes of intracellular GSH concentrations in single cells can be generated by dividing the fluorescence intensity values for the 405 nm channel by the 488 nm channel at each corresponding pixel (Fig. 4a, Supplementary Fig. 4, Supplementary Table 2). On treatment with a bolus of H$_2$O$_2$ (500 μM) to induce oxidative stress, the ratiometric imaging of HeLa cells revealed a decrease in the GSH level from 5.0 to 4.1 mM within 90 s (Fig. 4b,c). Subsequently, GSH ester (100 μM) was added to replenish the intracellular GSH loss, which led to recovery of GSH to the basal level in the next 3 min, as recorded by RT (Fig. 4b,c). The GSH-ester concentration refers to the concentration in the culture medium. It should be noted that GSH ester tends to be enriched inside cells due to its hydrophobic nature and esterase-catalyzed hydrolysis of GSH ester to GSH. Additionally, the intracellular GSH level changes measured using RT were confirmed by cell lysate-based liquid chromatography–mass spectrometry (LC–MS) measurements (Fig. 4d). These experiments demonstrate that RT reversibly reacts with GSH inside cells and can reflect intracellular GSH dynamics.

Neural activity results in increased production of ROS which have been linked to neurological disorders[20–22]. Therefore, sufficient GSH supply and active glutathione peroxidase are main defence resources of neurons under pro-oxidative conditions[23–25]. Recently, neural activity mediated by NMDA-receptors (NMDAR) has been coupled with transcriptional activation of glutamate-cysteine ligase, a key enzyme in GSH production, leading to a sustained GSH-based protection against an oxidative insult[26]. Here we tested whether NMDAR activation have a similar effect in neurons derived from human embryonic stem cells (ESCs; Supplementary Figs 5–7), a valuable and well established approach in modelling human neurological disorders in vitro[22,27]. By using patch-clamp, we first showed that derived neurons exhibited voltage-gated sodium-channel mediated action potentials and received synaptic inputs (Fig. 5a,b). Blocking GABA$_A$ receptors enhanced network activity and resulted in an increase in glutamatergic excitatory inputs (Fig. 5b). Finally, we used RT probe to monitor GSH levels in live neurons on transiently induced oxidative stress. As expected, the GSH levels were reduced by 100 μM of hydrogen peroxide but enhancing network activity with blockers of GABA$_A$ receptors (bicuculine) and KCNA channels (4-AP) attenuated this effect. Furthermore, we provide first data that the GSH level in human neurons may be mediated by NMDAR activation since the protective effect of enhanced network activity could be blocked with an NMDAR antagonist APV, consistent with the previous report (Fig. 5c,d)[26].

In addition, we applied flow cytometry to demonstrate that RT can be used for high-throughput quantification of GSH levels in single cells. HeLa cells were treated with a series of concentrations

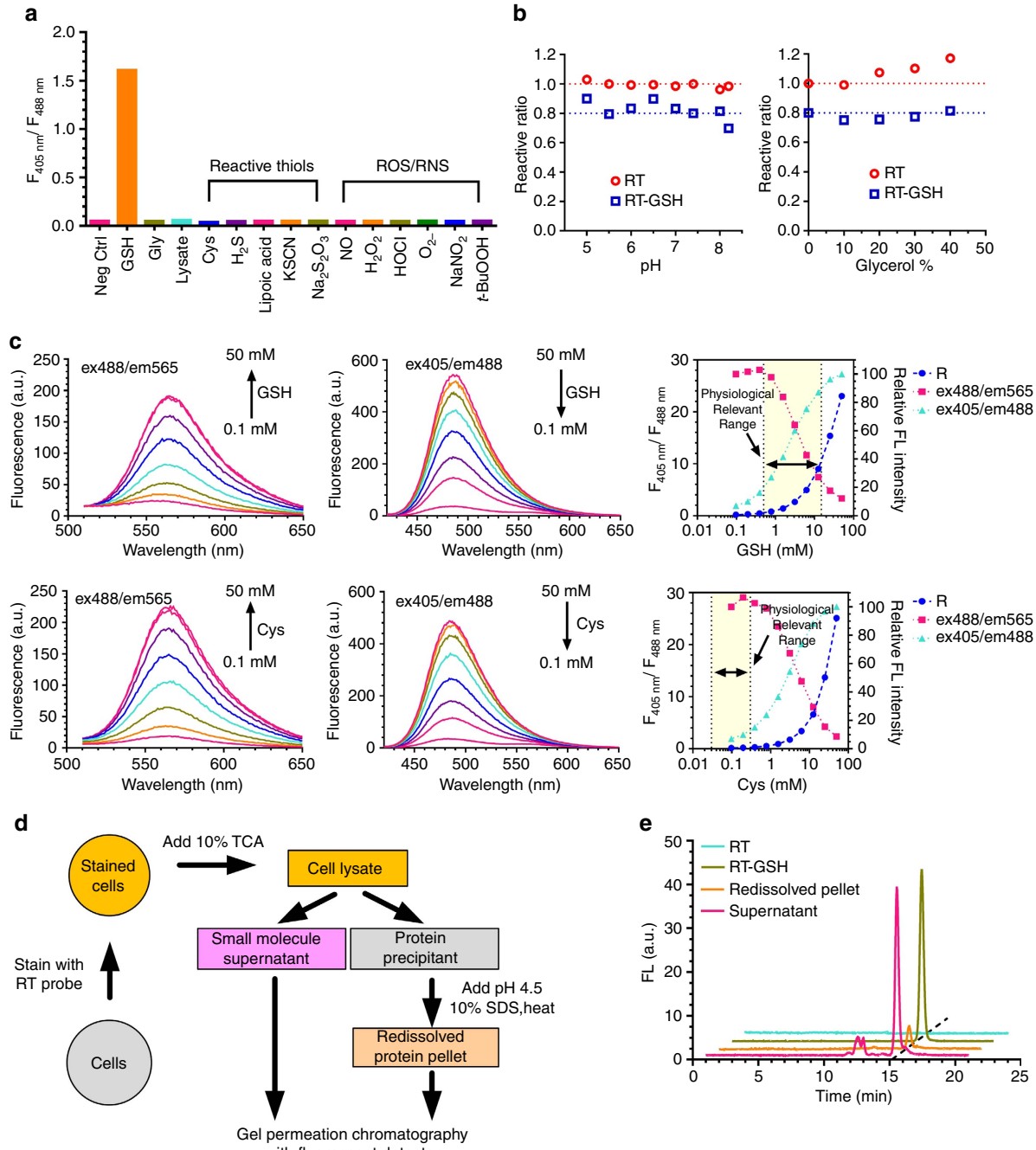

**Figure 2 | Reaction selectivity of RT towards GSH under different conditions.** (**a**) Fluorescence ratio after RT probe equilibrated *in vitro* with physiological relevant concentrations of different compounds. In particular, small-molecule thiols are removed in lysate for testing RT reactivity towards proteins *in vitro*. Refer to method for details. (**b**) Relative ratio of RT and RT-GSH adduct under different pH and viscosity conditions. Blue symbols are intentionally lowered by 0.2 units for clarity. (**c**) Responses of RT with GSH and cysteine. Fluorescence spectra of RT reaction with increasing concentrations of GSH and cysteine (0.1–50 mM) at $\lambda_{ex} = 488$ nm and $\lambda_{ex} = 405$ nm; maximum fluorescence signals and the corresponding ratio changes as a function of thiol concentrations are illustrated. Physiological concentrations of both thiols are shaded. (**d**) Flow chart of RT selectivity profiling against intracellular proteins in living cells. (**e**) Fluorescence signals ($\lambda_{ex} = 405$ nm, $\lambda_{em} = 478$ nm) of GPC traces of the lysate from RT stained HeLa cells. Each data point represents the mean value of three independent experiments. Error bars represent s.e.m. and are too small to show.

of buthionine sulfoximine (BSO) for 72 h, a specific inhibitor for the enzyme required in the first step of GSH synthesis (Fig. 6a). The GSH levels of the BSO-treated cells quantified using our RT-based flow cytometry method is well correlated to the values measured using cell lysates with a glutathione reductase-based biochemical assay ($R^2 = 0.99$, Fig. 6b).

Ferroptosis, a recently discovered iron-dependent form of nonapoptotic cell death, has been associated with

neurodegeneration and p53-mediated tumour suppression[28,29]. Ferroptosis is known to reduce the intracellular GSH level, but the dynamics of this process remains unexplored. In accordance with previous studies[28,29], we administered a ferroptosis inducer, erastin, to HT1080 cell culture and observed decrease in GSH levels by flow cytometry of RT signals (Fig. 7a,b). Interestingly, we discovered that such decreases in GSH levels during ferroptosis do not occur immediately on erastin treatment, despite dramatic

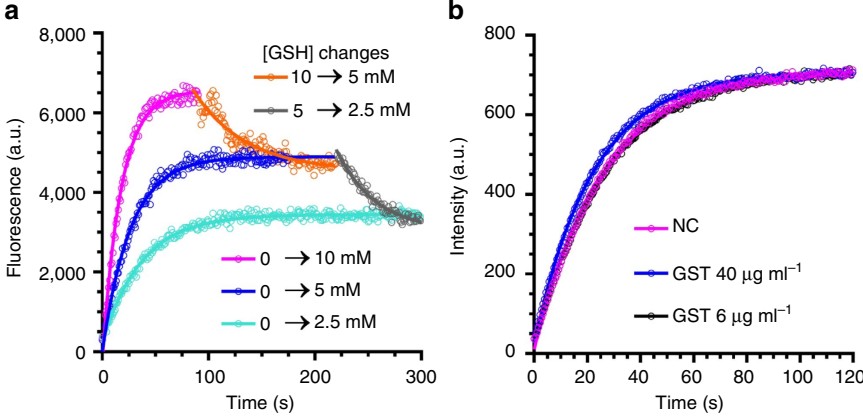

**Figure 3 | Kinetics of RT reaction with GSH. (a)** RT reacts rapidly with GSH in both forward and reverse reactions. All data were recorded at 37 °C. The kinetic traces for reverse reactions were shifted on x axis for clarity. **(b)** Catalytic effect of GST on the reaction rate between GSH and RT. Data points represent the fluorescence signal from newly generated RT-GSH conjugate. No significant differences can be observed with or without GST. One representative set of data out of three independent experiments are shown.

**Table 2 | Forward reaction rate constants between GSH probes and GSH.**

| GSH probe | Forward reaction rate constant $(M^{-1}s^{-1})$ | Reaction rate relative to TQG |
|---|---|---|
| TQG | 0.15 | 1 |
| RT | 7.5 | 50 |
| TQG-CN | 38.9 | 259 |
| RT-NH$_2$ | 29.1 | 194 |

changes in cellular morphology (Fig. 7c). This experiment demonstrates the convenience of using RT for high-throughput quantification of GSH levels in single cells and the power of using RT to monitor GSH levels during the initial phase of ferroptosis.

## Discussion

Michael addition reaction has been widely applied in organic synthesis and bioconjugation to build carbon–carbon and carbon–heteroatom bonds based on its forward reaction. However, the reverse reaction of Michael addition is under-utilized. Taking advantage of the reversibility of Michael addition reaction, we developed the first reversible reaction-based fluorescent probe ThiolQuant Green to perform single-point quantification of GSH in living cells[15]. Additionally, based on density functional theory, we developed a computational model to predict the $K_d$ values and the trend of reaction kinetics of Michael addition reactions[30]. Our extensive understanding of the physical organic chemistry of Michael addition reactions leads to the second generation RT GSH probe. Introducing an electron-withdrawing cyano group at the α position of the Michael acceptor of ThiolQuant Green can accelerate reaction kinetics but simultaneously shift the $K_d$ value into the μM range. Therefore, we replaced the ketone in ThiolQuant Green with a less reactive amide in RT to balance the $K_d$ back into the mM range. The α-cyano amide-based Michael acceptors have also been elegantly used by the Taunton group as warheads in small-molecule inhibitors to react with active site cysteines while being minimally affected by the high concentration of intracellular GSH due to the fast reaction kinetics and the mM of $K_d$ (refs 31,32). Furthermore, dialkylamino-coumarin is a class of dye known for their low quantum yields. Therefore, relatively high laser power is required for confocal imaging, which can consequently cause phototoxicity to live cells, especially with excitation at 405 nm (ref. 33). The

Lavis group made a seminal discovery that substituting the dialkylamino group with an azetidine group is a general method to significantly improve quantum yields of fluorophores[17]. Following Lavis's work, we found that introducing an azetidine group to RT can boost its quantum yield to 2.9%, which is threefolds of that of ThiolQuant Green. The quantum yield of RT-GSH is as high as 86%, which is 146-fold of that of ThiolQuant Green (Table 1). Additionally, the two carboxylic acid groups in RT are very important to achieve homogenous probe distribution in cells. The RT prototypes without the carboxylic acid groups, such as RT-NH$_2$ and TQG-CN, are fairly hydrophobic and cannot distribute in the nucleus, which is a hallmark of protein binding for fluorescent probes (Fig. 8a,b). The same inhomogeneous cellular distribution was also observed for ThiolQuant Green[15]. Interestingly, introduction of carboxylic acid groups to RT slows down its reaction kinetics towards GSH comparing to RT-NH$_2$ and TQG-CN (Table 2), presumably due to the charge repulsion between the carboxylic acid groups and the anionic GSH. Because inhomogeneous cellular distribution of reacted and unreacted probes may bias the ratiometric calculations, especially at the subcellular resolution, we opted for RT instead of RT-NH$_2$ to achieve high-quality quantitative results in living cells. We are currently exploring cationic or neutral water solubilizing groups to replace the carboxylic acids in RT to achieve higher temporal resolution while maintaining homogeneous cellular distribution, which will be reported in due course.

Through a series of optimization, we developed RT that enables quantitative real-time imaging of GSH in living cells. We demonstrated that RT preferentially reacts with GSH under physiological conditions and responds to both increases and decreases in GSH levels within a minute. Furthermore, RT has a high quantum yield and photostability. In addition, RT is capable of monitoring GSH changes on redox perturbance, which is essential in studying redox biology, especially fast processes, in living cells. Not only suitable for confocal imaging studies, RT can also be conveniently applied in flow cytometry to compare GSH levels and potentially be multiplexed with other probes and cell surface markers. It is worth noting that the Han group pioneered using small-molecule fluorescent probes to monitor the reversible redox cycles between peroxynitrite and GSH[34]. Unfortunately, unlike roGFPs measuring the redox potentials of GSH/GSSH, peroxynitrite and GSH are not a redox pair and the biological meaning of the ratio of these two species is elusive. In fact, Han's probe could be potentially used as a peroxynitrite probe, similar

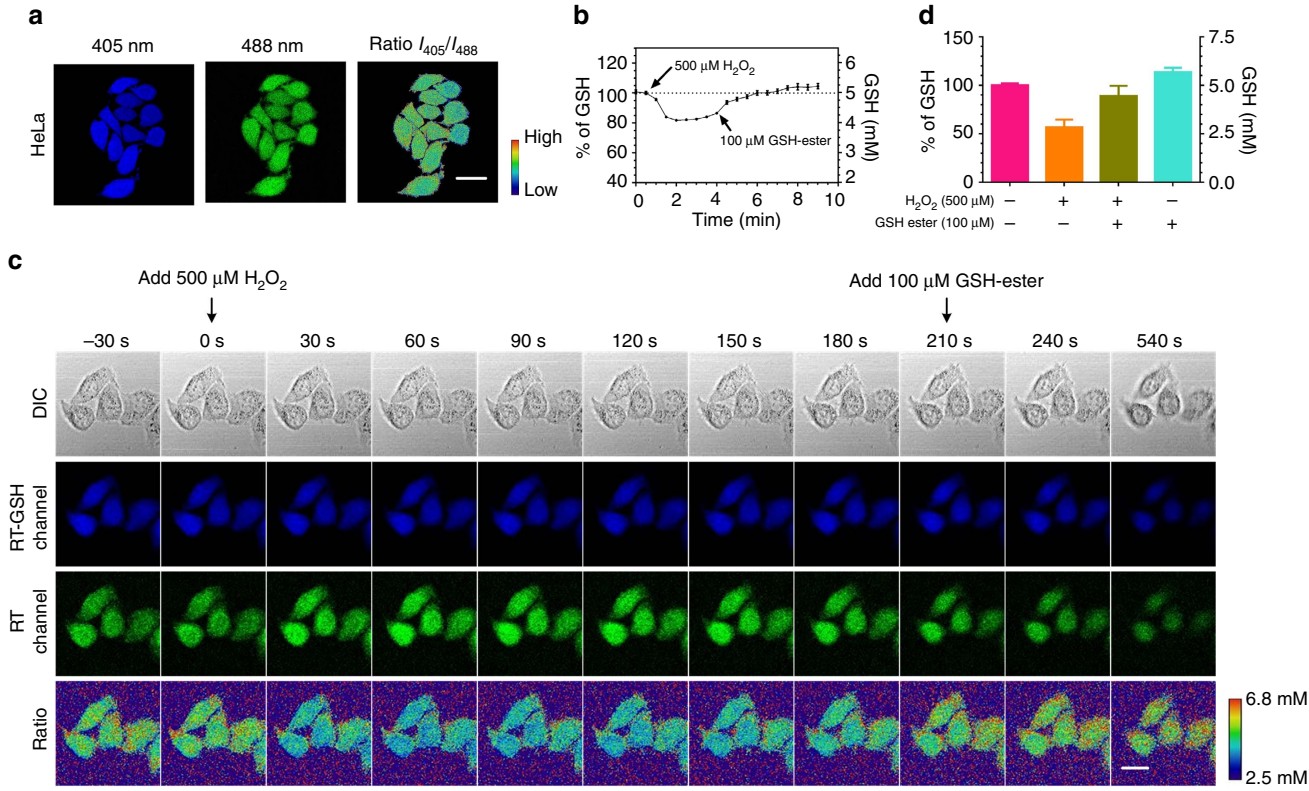

**Figure 4 | RT-based imaging and quantification of glutathione in living cells. (a)** Confocal and ratiometric images of HeLa cells stained with RT (refer to Supplementary Fig. 4 for details on generating ratiometric images). Scale bar, 20 µm. **(b)** Quantitative analysis of GSH level fluctuations in HeLa cells on consecutive treatment with $H_2O_2$ (500 µM) and GSH ester (100 µM). Each data point represents the mean value of 46 cells analysed from one representative time-lapsed imaging experiment. Error bars represent s.e.m. **(c)** Representative images of dynamic changes of GSH levels in HeLa cells on consecutive treatments with $H_2O_2$ and GSH ester. Scale bar, 20 µm. **(d)** Lysate-based GSH levels measured using LCMS. HeLa cells were treated with $H_2O_2$ only (orange), $H_2O_2$ followed by GSH ester (moss), GSH ester only (turquoise) or PBS (pink). The cells were lysed and the amounts of GSH were quantified using LCMS. Each data point represents the mean value of two independent experiments. Error bars represent s.d.

to the $H_2O_2$ probe Hyper, which is reversibly oxidized by $H_2O_2$ and reduced by thioredoxin[35]. Along similar lines, the biological meaning of many other small-molecule-based redox probes may also need careful interpretations[36,37]. During the manuscript submission process, two other fluorescent probes were independently reported to monitor GSH dynamics in living cells[38,39]. The Urano group reported the QuicGSH probe, elegantly taking advantage of the fast reversible reactions between GSH and a cationic silicon-rhodamine scaffold[38]. Unfortunately, due to the cationic nature of QuicGSH, it primarily accumulates in mitochondria. As the Urano group pointed out, using QuicGSH to monitor the global GSH dynamics requires a bold assumption that the mitochondrial and global GSH are at similar levels. However, it is well known that mitochondrial and cytosolic GSH levels are decoupled during cell death and many other biological processes[40]. The Yoon group also recently reported the QG-1 probe, which is structurally similar to our RT prototype TQG-CN (Fig. 8a) and suffers from inhomogeneous cellular distribution[39]. Our RT probe uniformly distributes inside cells. Therefore, RT is a fluorescent probe that can monitor global GSH dynamics in living cells. We envision that this new GSH probe will enable unprecedented opportunities to study GSH dynamics and transportation and revolutionize our understanding of the physiological and pathological roles of GSH in living cells.

## Methods

**Materials.** All the chemicals were purchased from Sigma-Aldrich or Alfa Aesar unless otherwise specified. 3-Carbaldehyde-7-azetidinylcoumarin (1) was

purchased from Ascendex Scientific, LLC. Tetrahydrofuran was distilled over sodium benzophenone ketyl and $CH_2Cl_2$ was distilled over phosphorus pentoxide. All the other solvents and reagents were used as received without further purification.

**Instruments.** Cary 60 UV–vis Spectrometer; Cary Eclipse Fluorescence Spectrophotometer; BioTek Synergy H1 Plate Reader; Carl Zeiss LSM 780 Confocal Microscope (from Optical Imaging and Vital Microscopy Core at Baylor College of Medicine); Teledyne ISCO CombiFlash Rf 200 flash liquid chromatography; Varian NMR ([1]H at 400 MHz); Bruker NMR ([1]H at 800 MHz); Agilent 6130 Single Quadrupole LC–MS; BD LSR II analysers (from Cytometry and Cell Sorting Core at Baylor College of Medicine).

**Software.** Fiji ImageJ[41,42]; FlowJo v.10.1; Prism Graphpad v.5; Zeiss Zen 2012 (blue edition) v.1.1.2.0.

**Solution preparation.** RT-AM stock solution was made by dissolving 3 mg of RT-AM in 1 ml of DMSO to obtain a final concentration of 5 mM. Stock solution can be kept at −80 °C for up to 1 year without any decomposition (verified by LC–MS). All testing solutions were prepared by diluting stock solution with DMSO and PBS/imaging buffer/culture medium.

Mixing was usually done by adding analyte solution (for example, GSH solution) into probe solution. If a surfactant is used, the surfactant should be mixed with the probe stock solution before performing any dilutions.

Cell staining solution was made by diluting 100 µM of RT probe in DMSO with a suitable imaging buffer for a final probe concentration of 1 µM. DMEM was used as the imaging buffer in this contribution if not otherwise specified.

**General cell culture and imaging.** The HeLa cell line used in this study was purchased from American Type Culture Collection (ATCC) and grown in DMEM (Gibco, 11965) media supplemented with 10% FBS and 1% 1003 Pen Strep (Gibco). Cells were cultured under a controlled atmosphere (37 °C, 5% $CO_2$). Glass bottom dishes were used for cell culture due to confocal scanning requirements. Cells were

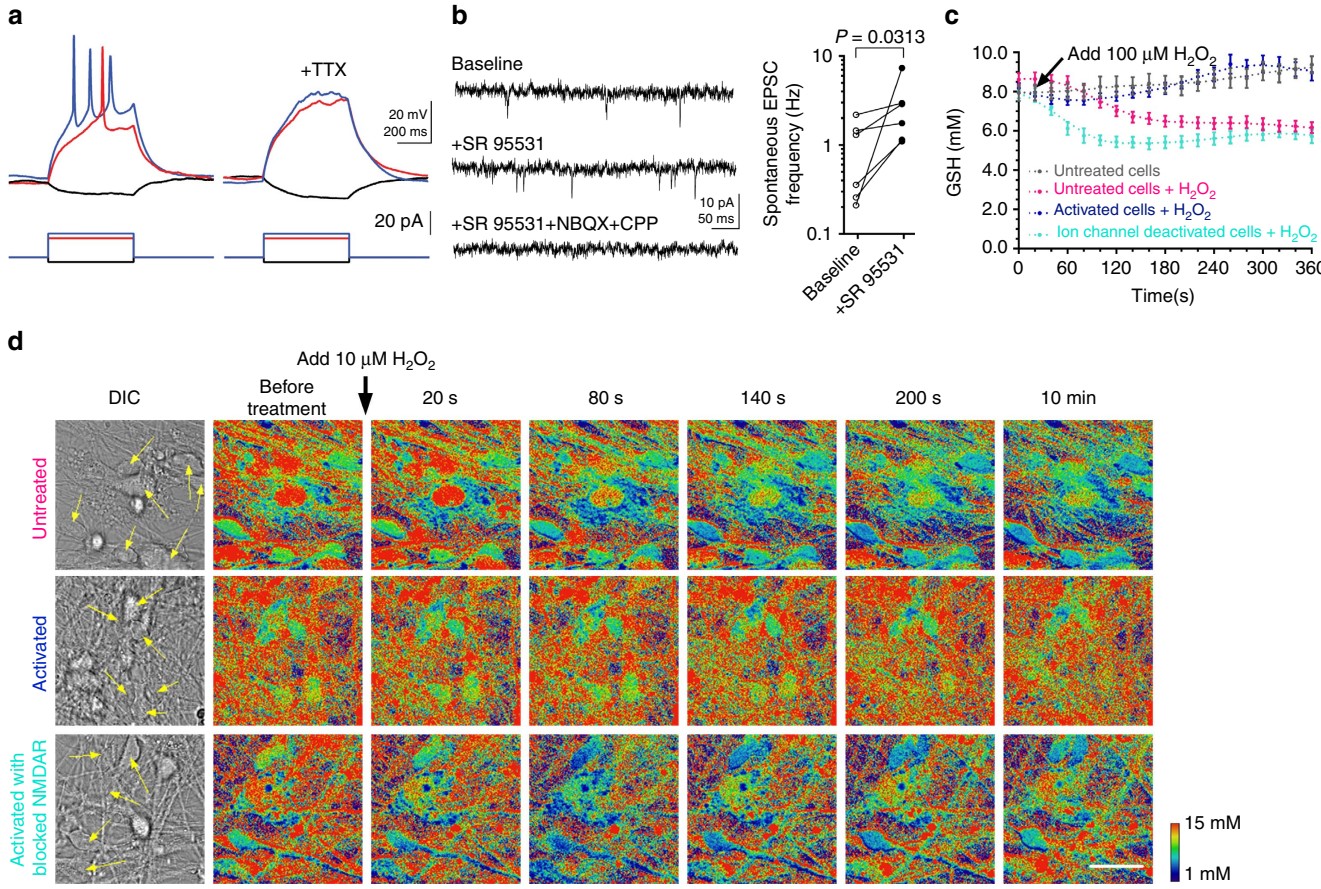

**Figure 5 | Neuron activity characterization and RT-based neuron imaging.** (**a**) Representative traces of membrane potentials (top panels) in response to somatic current injections (bottom panels) from a human ESC-derived neuron show action potentials, which were blocked by a voltage-gated sodium channel blocker, TTX (0.5 μM). Action potentials were detected in all 6 recorded neurons. The resting membrane potential, input resistance and membrane capacitance were $-64.3 \pm 1.5$ mV, $1.30 \pm 0.14$ GΩ and $33.9 \pm 8.1$ pF ($n = 6$), respectively. TTX was applied to three recorded neurons and blocked action potentials in all three neurons. (**b**) Representative traces of membrane currents recorded at a holding potential of $-70$ mV from a human ESC-derived neuron show spontaneous EPSCs in the presence or absence of a GABA$_A$ receptor antagonist SR95531 (10 μM), which were blocked by glutamate receptor antagonists, NBQX (10 μM) and CPP (10 μM). Spontaneous EPSC frequency was increased with SR95531 treatment in all 6 recorded neurons ($P = 0.0313$, $n = 6$, Wilcoxon matched-pairs signed rank test (two-sided)). NBQX and CPP were applied to 2 recorded neurons and blocked spontaneous EPSCs in both neurons. (**c**) Quantification of the real-time GSH levels in neurons treated with H$_2$O$_2$ in **d** (refer to Supplementary Fig. 2 for calibration curve). Each data point represents the mean value of 11 (grey), 16 (pink), 16 (blue) and 11 (turquoise) cells analysed from one representative time-lapsed experiment. Error bars represent s.e.m. (**d**) Time-lapsed ratiometric GSH imaging of untreated control (resting condition), bicuculline/4-AP treated (activation condition) and bicuculline/4-AP/APV treated (activation with paired inhibition of NMDAR channels) neurons on treatment with exogenous H$_2$O$_2$ (100 μM). The neural cell bodies are pointed out with arrows in the DIC images. Scale bar, 10 μm.

treated with RT-AM (1 μM with 1% DMSO in DMEM) for 10–15 min before imaging. Confocal images were acquired with 405 nm laser/418–495 nm filter and 488 nm laser/499–615 nm filter. All the microscope settings were kept consistent in each experiment.

For imaging, all cells were incubated in staining solution at room temperature and then moved to a 37 °C incubator. Two channels were used. For the best performance, typical imaging time should be <15 min; otherwise, anion transporter inhibitor, such as probenecid, should be used to prevent probe clearance.

The Grx1-roGFP2 HeLa cells used in this study were kindly provided by Dr Ninghui Cheng at Baylor College of Medicine. The Grx1-roGFP2 plasmid was kindly shared by Dr Tobias Dick. Confocal images were acquired with 405 nm laser/499–552 nm filter and 488 nm laser/499–553 nm filter.

Depending on the settings of cell imaging (including magnification, resolution, imaging length, imaging speed, laser power and stimulation method), the number of cells available for analysis varied from experiments. To ensure data quality and reproducibility, at least two biological replicates with >6 cells were analysed for each imaging experiment, and all the imaging experiments were repeated at least twice with one set of representative data shown in each figure. Specific statistics can be found in the corresponding figure legends.

**Differentiation and culturing of neural progenitor cell.** WA09 (H9) human ESCs were maintained on matrigel-coated plates in E8 media[43] (all media and

components are listed in Supplementary Tables 3 and 4). To differentiate ESCs into neural progenitor cells (NPCs), we used a variation of the dual SMAD inhibition protocol[44–49]. As depicted in our graphical abstract (Supplementary Fig. 5), ESCs were first dissociated with Accutase and 2 million cells dispensed per well of Aggrewell plate in neural induction medium to form aggregates. During the initial 24 h of culturing in Aggrewells we used 10 μM Y-27632 to promote cell survival. On the following day, $\frac{3}{4}$ of the media was changed and SMAD inhibition was initiated by the addition of 10 μM SB-431542. Dorsomorphin was administered at 4 μM concentration from day 3. At day 5 aggregates were gently collected, sieved through a reversible strainer and transferred into matrigel-coated plates with neural proliferation medium. From day 6 we included 10 μM cyclopamine to promote dorsalization. Both SMAD inhibitors and cyclopamine were present in the media until day 9. We changed media daily until rosette-shaped clusters of neural progenitors were harvested with rosette selection reagent between days 12 and 14. After dislodging, rosettes were incubated in wells coated with 0.2% porcine gelatin to allow the non-neural cells to differentially attach[21]. Floating fraction was collected after 1 h, transferred into non-coated cell-culture flasks and incubated in suspension overnight. On the following day, rounded and floating spheres of NPCs were plated into matrigel-coated 6-wells and propagated until confluency. From this point on, we used Accutase solution to make single-cell suspension for further expansion of NPCs or terminal differentiation into neurons. All cultures were maintained in the presence of 1% penicillin- streptomycin. B-27 supplement was vitamin A free.

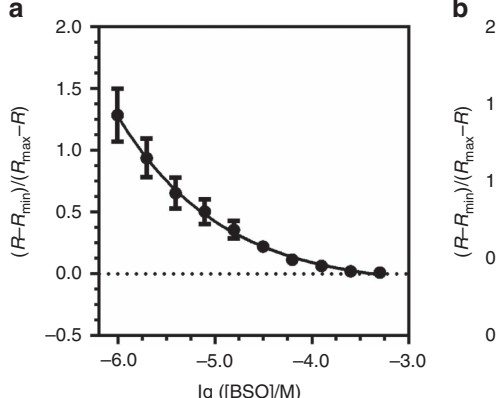
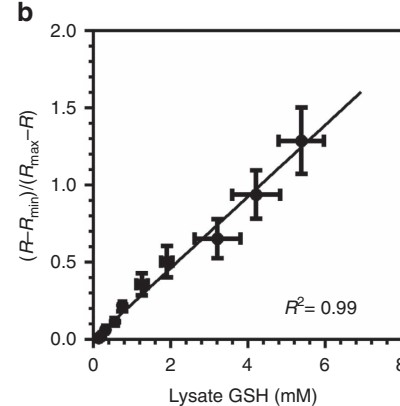

**Figure 6 | Comparison of GSH quantifications using RT probe and lysate-based measurements.** (**a**) A BSO concentration-dependent decrease of GSH levels measured using FACS. HeLa cells treated with a series of concentrations (0.9–500 µM) of BSO for 72 h were harvested and analysed using RT. $R = F_{405\,nm}/F_{488\,nm}$; $R_{min}$ and $R_{max}$ are the corresponding $R$ values with 0 and saturating GSH concentrations, respectively. Each data point represents the mean value of 17,000–28,000 cells analysed from three independent experiments. Error bars represent s.d. (**b**) Correlation of GSH levels in BSO-treated HeLa cells measured using the RT-based flow cytometry method in **a** and a lysate-based biochemical assay. Each point in the lysate measurements represents the mean value of a total of nine replicates from three independent experiments. Error bars represent s.d.

**Differentiation and culturing of human neurons.** Neural differentiation of NPCs was initiated by seeding $25 \times 10^3$ cells per square centimetre of matrigel-coated wells in neural proliferation medium. From the next day media was switched to neural differentiation medium and we kept feeding cells every 2 days. Starting from D17 cultured cells were maintained in neurobasal media without neurotrophins. Around D20 neurons were dissociated with Accutase and plated on matrigel-coated glass cover slips or in 35-mm culture wells with glass bottom. We maintained these cells in culture for up to 2 months until assays were performed. Media was changed once every 2–3 days and supplemented with $1 \mu g\,ml^{-1}$ laminin weekly to promote cell attachment. For this study, neurons have been differentiated twice from the frozen stock of NPCs.

**Immunocytochemistry.** Immunolabeling of NPCs and derived neurons was accomplished by utilization of a standard two-day protocol with: 15 min fixation in 4% PFA, 10 min permeabilization in 0.1% triton-PBS and 30 min blocking with 5% donkey serum. Following dilutions of antihuman primary antibodies in PBS were applied overnight at 4 °C: Nestin (Millipore, MAB5326, 1:200), SOX2 (Abcam, ab97959, 1:200), Doublecortin (Cell Signaling, 4604S, 1:200), MAP2 (Millipore, MAB3418, 1:200), PSA-NCAM (Millipore, MAB5324, 1:200), PSD95 (Abcam, ab18258, 1:200) Tubulin, beta 3 (Millipore, MAB1637, 1:200). For fluorescence staining, we incubated samples for 2 h with Alexa-488 and Alexa-594 conjugated donkey antibodies (1:200) against mouse and rabbit immunoglobulins (Supplementary Figs 6 and 7 are representative images from two sets of immunolabelled cultures of NPCs and neurons).

**Quantitative RT-PCR.** RNA isolation from human ESC and NPC cultures was done with the Aurum kit (Bio-Rad). We utilized Superscript III reverse transcriptase enzyme (Invitrogen) with random hexamer primers and 3 µg of RNA from each sample to produce cDNA according to the manufacturer's protocol. The cDNA was then fourfold diluted and used as a template in quantitative PCR with IQ SYBR Green Supermix (Bio-Rad) on a CFX96 instrument (Bio-Rad) using a 3-step protocol. All qRT-PCR samples were run in technical duplicates and the $\Delta\Delta Ct$ method based on the reference value estimate from 3 housekeeping gene (GAPDH, GUSB and TBP) was used to calculate relative expression level of each gene and reaction. Fold changes are finally given as the average fold change of replicates with the error bars which represent s.d. Plots on the Supplementary Fig. 6d are calculated from one of two independently performed RT-PCR runs. All used primers are listed in Supplementary Table 5.

**Electrophysiology.** Neurons were constantly perfused at 3 ml min$^{-1}$ with heated (30–32 °C), gassed (95% $O_2$, 5% $CO_2$) artificial cerebrospinal fluid (pH 7.4, 300–305 mOsm) containing (in mM): 119 NaCl, 2.5 KCl, 1.3 MgCl$_2$, 2.5 CaCl$_2$, 26 NaHCO$_3$, 1.3 NaH$_2$PO$_4$, 20 D-glucose and 0.5 sodium ascorbate. Recording pipettes were filled with internal solution (pH 7.34, 290 mOsm) containing (in mM): 142 potassium gluconate, 10 HEPES, 1 EGTA, 2.5 MgCl$_2$, 4 ATP-Mg, 0.3 GTP-Na and 10 Na$_2$-phosphocreatine. Neurons were visualized with infrared differential interference contrast imaging and a CCD camera under a water-immersion objective (40 ×, 0.8 numerical aperture). In whole-cell current clamp experiments, hyperpolarizing or depolarizing current pulses (500 ms, from − 10 to 30 pA in 5-pA steps) were injected into neurons to determine input resistances or trigger action potentials. In whole-cell voltage clamp experiments, excitatory postsynaptic

currents (EPSCs) were measured at the membrane potential of − 70 mV for 3 min. Voltage-gated sodium channels were blocked by 0.5 µM tetrodotoxin (TTX) (Tocris). GABA$_A$ receptors were blocked by 10 µM SR95531 (Tocris). $N$-Methyl-D-aspartic acid (NMDA) is α-amino-3-hydroxy-5-methyl-4-isoxazolepropionic acid (AMPA) were blocked by 10 µM (RS)-CPP (Tocris) and 10 µM NBQX (Tocris), respectively.

Data were low-pass filtered at 4 kHz and acquired at 10 kHz with an Axon Multiclamp 700B amplifier and an Axon Digidata 1550 Data Acquisition System under the control of Clampex 10.5 (Molecular Devices). Membrane potentials were not corrected for liquid junction potential. Data were analysed offline using AxoGraph X (AxoGraph Scientific). A scaled sliding template method (AxoGraph X) was used to detect spontaneous EPSCs. Data were low-pass filtered at 2 kHz and a 5-ms template (2 ms baseline, − 2 pA amplitude, 0.5 ms rise time, and 1 ms decay time) was used to detect spontaneous EPSCs. The detection threshold was set to 3 times the noise s.d. Events with amplitude < 5 pA were excluded. Statistical analyses were performed with Prism 6 (Fig. 5a,b).

**Neural activity tests.** Matured neuronal cultures in 35-mm culture wells with glass bottom were used to test whether prior neuronal activation and NMDAR signalling contribute to the GSH level in human derived neurons. To this end 24 h before RT imaging we started preconditioning neurons with culturing media which contained either 50 µM bicuculline and 250 µM 4-AP (conditioned for neuronal network activation) or 50 µM bicuculline, 250 µM 4-AP and 20 µM APV (conditioned for neuronal network activation with simultaneous blocking of NMDA receptor). Neurons were than loaded with RT probe and imaging experiments were performed as described earlier. All blockers were ordered from Sigma.

**Procedure for measurement on ultraviolet–visible and fluorimeter.** RT stock solution was diluted with PBS to the desired concentrations. Equal volumes (typically 1 ml) of various GSH solutions with different concentrations were mixed with the RT solution 10–15 min before measurement. All samples were first measured on ultraviolet–visible for absorption and then immediately transferred to the fluorimeter for fluorescent measurements. Two excitation wavelengths (405 and 488 nm) were used for fluorescent measurements. Photomultiplier gains were set to 600 and 700 for 405 and 488 nm excitations, respectively. For measuring kinetics, solutions were mixed inside the fluorimeter while data recording was on. Calculation of $K_d$ and $K_d'$ followed the same method as in the literature[15].

**Procedure for measurement on plate reader.** RT stock solution was diluted with PBS to the desired concentrations. Equal volumes (typically 20–100 µl) of various GSH solutions with different concentrations were mixed with the RT solution 10–15 min before measurement. All samples were prepared on one 96-well/384-well plate with 3 replicates each. The plate reader was set to read absorption of all samples first. Fluorescent signals were then recorded at $\lambda_{ex} = 405$ nm, $\lambda_{em} = 485$ nm, and $\lambda_{ex} = 488$ nm, $\lambda_{em} = 565$ nm with bottom read. Gain was set to 80 for both channels.

For testing fluorescent interference by environmental factors, the RT stock solution was directly diluted with corresponding solutions (for example, certain pH buffer or glycerol solution) and measured using the plate reader with the same settings stated above.

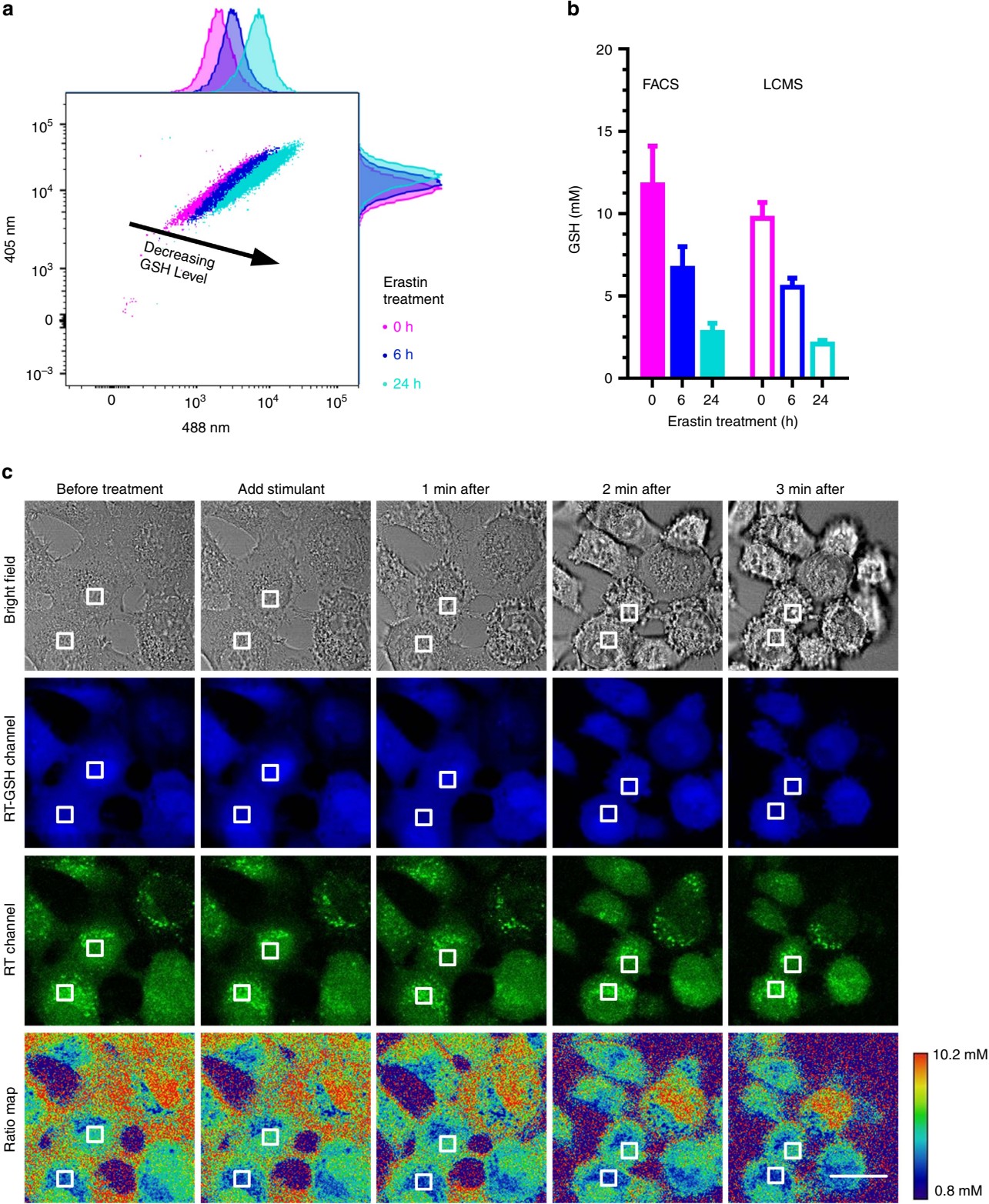

**Figure 7 | GSH quantification using RT-based FACS in ferroptotic HT1080 cells and representative images.** (**a**) Histogram of ferroptotic HT1080 cells measured using RT on a flow cytometer. Cells were treated with erastin (10 μM) for 0 (pink), 6 (blue) and 24 h (turquoise). (**b**) Statistical analysis of ferroptotic HT1080 cells measured with FACS and LCMS. For FACS, each data point represents the mean value of >13,000 cells analysed from two independent experiments. Standard parametric unpaired $t$-test was used to analyse the data with $P < 0.0001$ between groups. For LCMS, each data point represents the mean value of two independent experiments. Error bars represent s.d. (**c**) Representative images of HT1080 cells treated with 10 μM erastin. Scale bar, 10 μm. Despite significant morphology changes, GSH level did not change significantly over 3 min time span.

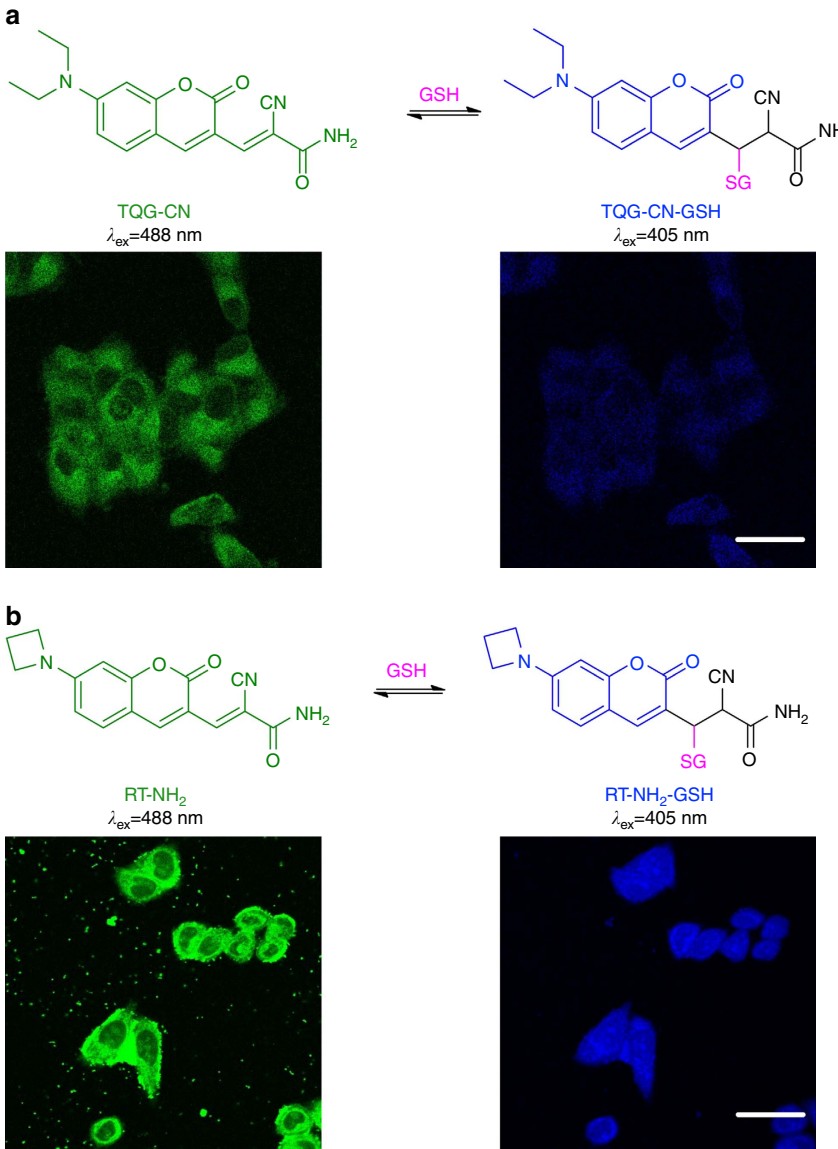

**Figure 8 | GSH probe prototypes.** (**a**) Chemical structures of TQG-CN and TQG-CN-GSH and its confocal images in HeLa cells. (**b**) Chemical structures of RT-NH$_2$ and RT-NH$_2$-GSH and its confocal images in HeLa cells. Scale bar 10 μm. It should be noted that neither TQG-CN nor RT-NH$_2$ distributes in the nucleus, which is a hallmark of protein binding for fluorescent probes.

**Procedure for RT probe selectivity tests with small molecules *in vitro*.** Selectivity was tested in a 384-well plate. RT solution was mixed with various compounds that could potentially interfere with signals. A panel of physiological relevant nucleophiles and reactive oxygen/nitrogen species (ROS/RNS) were selected. Little consumption of RT was observed in the presence of glycine, cysteine and ROS/RNS.

**Procedure for RT probe selectivity test with cell lysate *in vitro*.** To test if RT reacts or interacts with intracellular proteins, we obtained HeLa cell lysate by physically scratch off the cells from a culture dish in RIPA buffer. The mixture was then centrifuged at 15,000 g for 20 min at 4 °C and the supernatant was collected. Protein concentration was determined by Bradford assay to be 1.1 mg ml$^{-1}$. To eliminate small-molecule thiol (such as GSH) interference, we centrifuged the lysate with 3 K cut-off membrane at 7,500 g for 40 min at 4 °C three times. The volume of solution decreased from 2 ml to 250 μl after each spin, and 1.75 ml of fresh PBS was added before the next spin. After washing, the protein concentration was measured again to be 1.1 mg ml$^{-1}$, which implies >99% recovery of all proteins from the lysate. The resulting small-molecule free lysate was then concentrated twice to a concentration of 36 mg ml$^{-1}$, during which we lost <10% of total protein. The concentrated lysate was then mixed with RT solution and measured on a plate reader for fluorescence. No fluorescence ratio change can be observed after mixing RT with lysate.

**Procedure for RT probe selectivity test with cell lysate *ex vivo*.** Procedure was adopted from Hansen *et al.*[50] with minor optimization (refer to graphic abstract on Fig. 2d). HeLa cells were grown to 95% confluent on 10-cm dishes. Before analysis, cells were stained with 5 μM RT for 10 min at room temperature. After staining, cells were washed with cold PBS twice and lysed with 10% trichloroacetic acid on ice. After sonication, cell lysate was centrifuged at 12,000 g for 15 min at 4 °C and the resulting supernatant was directly analysed using GPC with a fluorescence detector (GPC-FL). The precipitated protein pellet was re-dissolved by adding pH 4.5 citrate (0.4 M) buffer with 5% SDS and 1 mM EDTA (the volume is the same as the supernatant), and heated to 45 °C for 5 min. After sonication, cell lysate was centrifuged at 12,000 g for 15 min at room temperature and the supernatant was analysed using GPC-FL. GPC running buffer was pH 4.5 citrate buffer (0.1 M) with 0.1% SDS and 1 mM of EDTA. Column temperature for GPC was 50 °C. Fluorescent detector was set to detect signals with $\lambda_{ex} = 405$ nm and $\lambda_{em} = 478$ nm. The integration results revealed that 10% of RT reacts with protein thiols and 90% of RT reacts with small-molecule thiols, presumably GSH.

**Quantification of GSH using FACS.** HeLa cells were cultured in 10% FBS DMEM containing different concentrations of BSO (0.9, 1.9, 3.9, 7.8, 15.6, 31.3, 62.5, 125, 250 and 500 μM) for 72 h and harvested for analysis. Cells were then suspended in fresh medium containing 1 μM of RT probe as single-cell suspensions. To minimize the potential clearance of RT from cells, we strictly kept the staining time between 9 and 10 min for each sample. Cells were then analysed with a FACS

analyser. PE and pacific-blue channels were selected to measure the fluorescence intensities of unreacted and reacted RT, respectively. A cell-only control was used to adjust the PMT for each channel. Live and dead cell discrimination was based on forward (FSC-A) and side scattering (SSC-A), and doublets were excluded based on FSC-A and FSC-H. Single cells were further gated based on SSC-A and SSC-H. Fluorescent signals from both channels that were over 4 s.d.'s were excluded from the final data set used for analysis. In each group, $>10,000$ cells were initially analysed using FACS. Three replicates were performed for each FACS experiment and $\sim17,000$–28,000 cells were eventually included in statistical analysis (Fig. 6a,b). $R = F_{405\,nm}/F_{488\,nm}$; $R_{min}$ and $R_{max}$ are defined as the corresponding $R$ values with 0 and saturating GSH concentrations, respectively. However, $R_{min}$ and $R_{max}$ are not measurable parameters on FACS because we cannot manipulate the intracellular GSH level to zero or saturating conditions. $R_{min}$ and $R_{max}$ were obtained by fitting the $R$ values and the lysate-based GSH concentrations $C_{GSH}$ using equation: $(R - R_{min})/(R_{max} - R) = k \times C_{GSH}$, in which $k$ is a constant. The linear fitting afforded $R_{min} = 1.65$, and $R_{max} = 2.98$.

For generating calibration curve of HT1080 cells, exact same conditions were used as described above. And fitting afforded $R_{min} = 1.60$, and $R_{max} = 6.01$.

HT1080 cells were treated with $10\,\mu M$ of erastin for 0, 6 and 24 h before harvesting. Cells were stained and analysed with a FACS analyser as stated above. In each group, 30,000 cells were initially analysed using FACS and $\sim21,000$–24,000 cells were eventually included in statistical analysis. Two replicates were performed for each FACS experiment with one set of representative data shown (Fig. 7b).

**Quantification of GSH using Ellman's assay.** Cells were harvested and lysed with 0.1% Triton-X and 0.6% sulfosalicylic acid in 0.1 M potassium phosphate buffer with 5 mM EDTA at pH 7.5 to afford protein free lysate. Typically, $10^6$ cells will be lysed to 1 ml of solution. In a 96-well microtiter plate, $20\,\mu l$ of cell lysate was prepared in each well. Freshly made solutions of equal volume 5,5'-dithio-bis(2-nitrobenzoic acid) and glutathione reductase with concentrations of $0.33\,g\,l^{-1}$ and 1.67 units per ml, respectively, were mixed with lysate to a final volume of $140\,\mu l$. The mixture was incubated at room temperature for 1 min. Then, $60\,\mu l$ of β-NADPH ($0.67\,g\,l^{-1}$) was added and the mixture was immediately measured for absorbance at 412 nm every 30 s for 2 min. The slope of the absorbance changes was proportional to the GSH concentration. A standard curve with known GSH concentrations was used to calibrate all the results. A more detailed protocol can be found in the literature[15]. Based on the amount of GSH in the lysate, we calculated the GSH concentration assuming an averaged HeLa/HT1080 cell volume of $4,000\,\mu m^3$ (refs 15,51,52).

**Quantification of GSH using LCMS.** Cells were incubated with $100\,\mu M$ of N-methyl maleimide (NMM) in PBS for 15 min at room temperature to derivatize all the small-molecule thiols. Then, the cells were harvested and lysed with 0.1% Triton-X and 0.6% sulfosalicylic acid in 0.1 M potassium phosphate buffer with 5 mM EDTA at pH 7.5 to afford a protein free lysate. Typically, a million cells were lysed in 1 ml of lysis buffer. The samples were diluted by 100 times and measured using LCMS in the selective ion mode. Two ions were monitored: GSH-NMM ($m/z$: 419.1, M + 1) and Cys-NMM ($m/z$: 233.0, M + 1). A standard curve with known concentrations of GSH-NMM and Cys-NMM were used to calibrate all the results.

To confirm the GSH levels changes measured using RT in Fig. 4c, HeLa cells were seeded into 6-well plates 24 h before experiment. For treatments, similar to our imaging conditions, cells were treated with $500\,\mu M$ $H_2O_2$ for 10 min; $500\,\mu M$ $H_2O_2$ for 10 min followed by $100\,\mu M$ GSH ethyl ester for 10 min; and $100\,\mu M$ GSH ethyl ester for 10 min. After treatments, cells were washed three times with PBS, and incubated with 1 mM NMM for 5–10 min. Wash with PBS one more time and add $500\,\mu l$ of lysis buffer (0.1% Triton-X and 0.6% sulfosalicylic acid in 0.1 M potassium phosphate buffer with 5 mM EDTA at pH 7.5) on ice. Collect the cell lysate and centrifuge at $13,200g$ at $4\,°C$ for 15 min, and the supernatants were used for LC–MS analysis after diluting with water to proper concentrations. Based on the amount of GSH in the lysate, we calculated the GSH concentration assuming an averaged HeLa cell volume of $4,000\,\mu m^3$ (refs 15,51,52).

**Data availability.** The data that support the findings of this study are available from the corresponding author upon reasonable request.

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

## Acknowledgements

The research was supported in part by the National Institutes of Health (R01-GM115622 and R01-CA207701 to J.W., R01-AG045183, R01-AT009050, R21-EB022302 and DP1-DK113644 to M.C.W., R01-GM120033 to M.M.-S., R01-NS100893 to M.X.), the Welch Foundation (Q-1912 to M.C.W.), Cancer Prevention and Research Institute of Texas (CPRIT R1104 to J.W., RP130573 to M.M.-S.), Whitehall Foundation Research Grant (2015-05-54 to M.X.), the Curtis Hankamer Basic Research Fund at Baylor College of Medicine (to M.X.), Caroline DeLuca Scholarship (to M.X.), the United States Department of Agriculture, Agricultural Research Service through Cooperative Agreement Number 58-6250-0-008 (to N.C.), the IDDRC Microscopy Core (P30HD024064 and 1U54 HD083092 Intellectual and Developmental Disabilities Research Grant from the Eunice Kennedy Shriver National Institute of Child Health and Human Development), the Optical Imaging and Vital Microscopy core, and the Cytometry and Cell Sorting Core at Baylor College of Medicine with funding from the NIH (AI036211, CA125123, and RR024574) and the expert assistance of Joel M. Sederstrom. Neural progenitor cells used in this study were derived from the WA09 human ESC line (H9 ESC) derived by Dr James Thomson and distributed by WiCell Research Institute under SLA agreement. We are thankful to Dr Jean Kim and the Human Stem Cell Core at Baylor College of Medicine for mediating the cell transfer, Dr Tobias Dick for Grx1-roGFP2 plasmid, and Drs Vsevolod Belousov, Luke Lavis, and Catherine Goodman for thoughtful discussions.

## Author contributions

X.J., J.C., A.B., C.Z., M.X., N.C., C.P.S., F.X., M.C.W., M.M.-S. and J.W. designed experiments; X.J., J.C., A.B., C.Z., X.S., S.L.C., Z.-L.C., M.T., F.L., K.R.M. and A.C.M.F. performed experiments; X.J., J.C., A.B., M.C.W., M.M.-S., and J.W. performed data analysis and wrote the manuscript.

## Additional information

**Competing interests:** X.J., J.C. and J.W. are co-inventors of a patent application related to this work. The remaining authors declare no competing financial interests.

DOI: 10.1038/ncomms16163

# Corrigendum: Quantitative real-time imaging of glutathione

Xiqian Jiang, Jianwei Chen, Aleksandar Bajić, Chengwei Zhang, Xianzhou Song, Shaina L. Carroll, Zhao-Lin Cai, Meiling Tang, Mingshan Xue, Ninghui Cheng, Christian P. Schaaf, Feng Li, Kevin R. MacKenzie, Allan Chris M. Ferreon, Fan Xia, Meng C. Wang, Mirjana Maletić-Savatić & Jin Wang

*Nature Communications* 8:16087 doi: 10.1038/ncomms16087 (2017); Published 13 Jul 2017; Updated 3 Oct 2017.

Previous work by Cho and Choi describing the development of a cyanoacrylamide-based fluorescence sensor for reversible detection of thiols in homogenous solutions was inadvertently omitted from the reference list of this Article. This work should have been cited in the first paragraph of the discussion, following the rationale behind the development of the Michael acceptor, as follows: 'A fluorescent sensor based on the cyanoacrylamide Michael acceptor has previously been shown to reversibly react with thiols in homogenous solutions but without any cellular applications, possibly due to the low quantum yield and poor aqueous solubility (Cho *et al.*, 2012)'.

Cho, A. Y. & Choi, K. A coumarin-based fluorescence sensor for the reversible detection of thiols. *Chem. Lett.* **41**, 1611–1612 (2012).

