## [Peer Review File · Nature Communications]

Reviewers' comments:

Reviewer #1 (Remarks to the Author):

The manuscript of Jiang et al. reports on the synthesis and biological use of a new probe for GSH determination in living cells. The results are very impressive and the probe will be of real help in studying GSH dynamics in cell. However, there are some concerns about the data presented.

1. The cysteine concentration used in Fig 1 is 100uM while the GSH is 10 mM. What if the cysteine is 10 mM? Cysteine in some conditions or cell type is present at high level.
2. Also the replenishment of GSH by using GSH-ester is not clear (Fig.1F). How is possible that by treating the cells with 100 uM GSH-ester the authors observed a change in GSH from 4 to 5 mM?
3. Figure S14. GSH Levels in the Early Phase of Ferroptosis Process. Why the authors selected the indicated cells? In the same panel there are cells in which the changes appeared more interesting (upper and right). If this is the real trend in the cell population how many cells and analyses need to be performed in order to have a more general phenomenon.
4. In the ferroptosis experiments which are the changes in GSH concentration determined by HPLC?

Reviewer #2 (Remarks to the Author):

The authors report a fast, reversible, and ratiometric fluorescent probe for real-time glutathione imaging in living cells. They report the synthesis and in vitro characterization of an azetidino-coumarin with optimized functionality for efficient photophysical properties, reaction kinetics, and cellular uptake. The response, kinetics, and selectivity versus small molecule thiols and reactive species (including potential interference with GST) are characterized in vitro. Selectivity versus protein thiols is demonstrated by GPC experiments of probe bound to small molecule thiols vs probe bound to proteins in cell lysates. Photostability and changes in GSH levels in response to peroxide or added GSH are measured in HeLa cells. The authors demonstrate that their probe can be used in conjunction with flow cytometry to measure reduction in GSH levels upon addition of the inhibitor BSO. Finally, they use their probe to monitor GSH levels in response to oxidative stress in neurons and to illuminate the dynamics of decreased GSH during ferroptosis in HT1080 cells.

This is a highly significant advance in that it achieves real-time quantification of small molecule species in living cells, a long-standing challenge in the field of reaction-based fluorescent probes. This work will be broadly appreciated by chemists, biologists, and imaging scientists. The experimental work is professionally executed and convincing. Extensive supporting data is included and details (including number of replicates and data analysis) are accurately reported, which will certainly help other researchers use this approach and effectively navigate any alternative interpretations of the data. The authors have anticipated and collected data to address selectivity versus protein thiols. Given the high quality and significance, I recommend that this study by accepted for publication.

While suitable for publication in its current form, I include the following suggestions for the authors to consider in strengthening their manuscript before final publication:

1. The manuscript text could better explain the experiment performed to collect the data in Figure 2b and 2c. What does bicuculline, 4-AP, and 4-APV do? This is not really clear from the text, and a casual reader may have trouble understanding this.
2. I recommend removing the explanatory statements from figure captions in the manuscript. The explanations in the SI figure captions are very helpful, but for the manuscript, it would be more appropriate to have these in the text and not the caption.

3. Maybe tone down language of "first" fluorescent probe for "global" GSH. This is clearly a highly significant advance and I don't think it needs to be over-sold in this way. The Urano paper is appropriately cited and the differences are accurately noted.

4. Including ^{13}C NMR and HRMS for final probe compound would assist in the reproducibility of the work.

5. The authors give an estimate for the cell volume in the SI. A reference or explanation for this estimate would be appropriate.

Reviewer #1 (Remarks to the Author):

The manuscript of Jiang et al. reports on the synthesis and biological use of a new probe for GSH determination in living cells. The results are very impressive and the probe will be of real help in studying GSH dynamics in cell. However, there are some concerns about the data presented.

1. The cysteine concentration used in Fig 1 is 100uM while the GSH is 10 mM. What if the cysteine is 10 mM? Cysteine in some conditions or cell type is present at high level.

We thank the reviewer for pointing out the potential interference from cysteine. In general, intracellular cysteine is presented at a concentration below 100 μM (*The Journal of biological chemistry* **287**, 4397-4402 (2012); *The FASEB journal* **18**, 1246-1248 (2004)). Excess intracellular cysteine has been demonstrated to be toxic in various animal models (*The Journal of nutrition* **136**, 1652S-1659S (2006)). Therefore, we chose the high end of physiologically relevant concentration of cysteine, which is 300 μM , to determine the potential interference of intracellular cysteine to glutathione in testing RT probe. Because we could not achieve cysteine concentration as high as 10 mM inside cells, we performed a test tube experiment to study the reaction between RT and cysteine. As expected, RT has similar reactivities towards cysteine comparing with GSH (Supplementary Fig. 5). We agree with the reviewer that cysteine may interfere with RT signals if cysteine is in the mM range. To the best of our knowledge, such high cysteine concentration has not been observed in mammalian cell lines. To be cautious, it is always a good practice to measure the GSH and cysteine levels using LC-MS in cell lysates to confirm the results measured using RT.

2. Also the replenishment of GSH by using GSH-ester is not clear (Fig.1F). How is possible that by treating the cells with 100 μM GSH-ester the authors observed a change in GSH from 4 to 5 mM?

We thanked the reviewer for pointing this out. We did not explain the condition well enough. In a typical replenishment experiment, we usually use 10^5 HeLa cells in 1 mL of medium. A typical HeLa cell volume is between 2000-4000 μm^3 , so that the total volume of 10^5 HeLa cells is $2-4 \times 10^8 \mu\text{m}^3$, which is $2-4 \times 10^{-4}$ mL. Considering that we have 100 μM of GSH-ester in 1 mL of medium, and that GSH-ester is lipophilic which tends to accumulate on cell surface, each cell could have access to 250-500 mM of GSH-ester. This is well above the maximum GSH-ester level that each cell can uptake. We actually saturated the cells with GSH under our experiment conditions, because we observed similar level of GSH increase with 100, 200 and 400 μM GSH-ester treatments to cells. We clarified the condition in updated manuscript by adding the following text "The GSH-ester concentration refers to the concentration in the culture medium. It should be noted that GSH-ester tends to be enriched inside cells due to its hydrophobic nature and esterase-catalyzed hydrolysis of GSH-ester to GSH. Additionally, further increasing the GSH-ester concentration to 400 μM led to the same intracellular GSH level, suggesting that cells may have a saturating level for GSH."

3. Figure S14. GSH Levels in the Early Phase of Ferroptosis Process. Why the authors selected the indicated cells? In the same panel there are cells in which the changes appeared more interesting (upper and right). If this is the real trend in the cell population how many cells and analyses need to be performed in order to have a more general phenomenon.

HT1080 cells were chosen based on the work from the Stockwell group on ferroptosis. HT1080 is highly sensitive to ferroptosis inducer erastin. Due to the low throughput of confocal imaging experiments, it would be difficult to perform statistically significant measurements purely based on confocal imaging. Because our method can be easily applied to a much higher throughput FACS settings, we added LCMS and FACS experiments to determine the GSH changes in both the early (Supplementary Fig.15) and late (Fig.2g) time points of ferroptosis. We did not observe a significant change in GSH levels from both measurement methods. However, LC-MS measurements require a very long processing time (typically >30 min), which caused some undesired errors in the measurement especially on the early time points. The statistical analysis of more than 13,000 cells at each corresponding time point should be sufficient to support our observations in the imaging experiment.

4. In the ferroptosis experiments which are the changes in GSH concentration determined by HPLC?

We thanked the reviewer for pointing this out. We added the LCMS measurement results in the revised manuscript (Fig. 2g and Supplementary Fig. 15). The LCMS results are consistent with our previous observations.

Reviewer #2 (Remarks to the Author):

The authors report a fast, reversible, and ratiometric fluorescent probe for real-time glutathione imaging in living cells. They report the synthesis and in vitro characterization of an azetidino-coumarin with optimized functionality for efficient photophysical properties, reaction kinetics, and cellular uptake. The response, kinetics, and selectivity versus small molecule thiols and reactive species (including potential interference with GST) are characterized in vitro. Selectivity versus protein thiols is demonstrated by GPC experiments of probe bound to small molecule thiols vs probe bound to proteins in cell lysates. Photostability and changes in GSH levels in response to peroxide or added GSH are measured in HeLa cells. The authors demonstrate that their probe can be used in conjunction with flow cytometry to measure reduction in GSH levels upon addition of the inhibitor BSO. Finally, they use their probe to monitor GSH levels in response to oxidative stress in neurons and to illuminate the dynamics of decreased GSH during ferroptosis in HT1080 cells.

This is a highly significant advance in that it achieves real-time quantification of small molecule species in living cells, a long-standing challenge in the field of reaction-based fluorescent probes. This work will be broadly appreciated by chemists, biologists, and imaging scientists. The experimental work is professionally executed and convincing. Extensive supporting data is included and details (including number of replicates and data analysis) are accurately reported, which will certainly help other researchers use this approach and effectively navigate any alternative interpretations of the data. The authors have anticipated and collected data to address selectivity versus protein thiols. Given the high quality and significance, I recommend that this study be accepted for publication.

While suitable for publication in its current form, I include the following suggestions for the authors to consider in strengthening their manuscript before final publication:

1. The manuscript text could better explain the experiment performed to collect the data in Figure 2b and 2c. What does bicuculline, 4-AP, and 4-APV do? This is not really clear from the text, and a casual reader may have trouble understanding this.

We thank the reviewer for the suggestion. We added more information on the detailed functions of these compounds in the revised manuscript. We added the following text “As expected, the GSH levels were reduced by 100 μ M of hydrogen peroxide but enhancing network activity with blockers of GABA_A receptors (bicuculline) and KCNA channels (4-AP) attenuated this effect. Furthermore, we provide first data that the GSH level in human neurons may be mediated by NMDAR activation since the protective effect of enhanced network activity could be blocked with an NMDAR antagonist APV, consistent with the previous report (Figs. 2b and 2c).”

2. I recommend removing the explanatory statements from figure captions in the manuscript. The explanations in the SI figure captions are very helpful, but for the manuscript, it would be more appropriate to have these in the text and not the caption.

We thank the reviewer for the suggestion. We moved the explanatory statements from figure caption to the main text.

3. Maybe tone down language of “first” fluorescent probe for “global” GSH. This is clearly a highly significant advance and I don’t think it needs to be over-sold in this way. The Urano paper is appropriately cited and the differences are accurately noted.

We take this as a compliment and thank the reviewer for recognizing the significance of our research. We changed the description of our probe from “the first probe” to “a probe” in the revised manuscript.

4. Including ¹³C NMR and HRMS for final probe compound would assist in the reproducibility of the work.

We added all the NMR and HRMS data for our final compound in the revised manuscript. The final product RT-AM was extensively characterized with ^1H , ^{13}C , ^{13}C -HSQC, ^{13}C -HMBC, and NOSEY NMR, and HRMS. Spectra can be found in the supporting information.

5. The authors give an estimate for the cell volume in the SI. A reference or explanation for this estimate would be appropriate.

A typical HeLa cell volume is between 2000–4000 μm^3 according to previous literatures. We also measured the cell volume by ourselves using PCV tubes in our previous publication, which is in the range of the reported value. We included these numbers and references in the revised manuscript.

Reviewers' comments:

Reviewer #1 (Remarks to the Author):

The revised manuscript is generally improved. However, I have problem with the response on point 2.

The authors give a very good mathematical exercise to finally declare that the cells experienced a very high concentration of GSH-ester, but I cannot agree with their conclusion. It is really more scientifically sound to demonstrated, with another experimental approach, the increase of intracellular GSH and to make the comparison in percentage of increment between the two methods employed.

Reviewer #2 (Remarks to the Author):

The authors have followed all of the suggested edits and I recommend publication in its current form.

Reviewer #1 (Remarks to the Author):

The revised manuscript is generally improved. However, I have problem with the response on point 2. The authors give a very good mathematical exercise to finally declare that the cells experienced a very high concentration of GSH-ester, but I cannot agree with their conclusion. It is really more scientifically sound to demonstrated, with another experimental approach, the increase of intracellular GSH and to make the comparison in percentage of increment between the two methods employed.

We thank the reviewer for pointing out the necessity to verify the RT based GSH quantification in Fig.1f using a secondary method, especially to explain why addition of 100 μ M of GSH ester can cause a mM change of the intracellular GSH level. We added a new experiment to measure the GSH level changes upon H₂O₂ and/or GSH ester treatments using LC-MS (Supplementary Fig.10b and the table below). We found that the lysate based LC-MS measurements are in agreement with the RT based measurements. It should be noted that the absolute GSH levels measured using LC-MS are slightly different from the RT based measurements. This may be because the lysate based measurements require extensive processing of cells, which may cause GSH level changes. Nonetheless, we showed that addition of 100 μ M of GSH ester can cause mM of increase of the GSH level in cells. **Please note that all the changes to the manuscript are highlighted in red.**

Treatment	LC-MS based GSH level (mM)	Note
Untreated	5.0	
H ₂ O ₂ only (500 μ M)	2.8	
H ₂ O ₂ (500 μ M) followed by GSH ester (100 μ M)	4.5	Compared to H ₂ O ₂ only, the GSH level increased by 1.7 mM with the addition of 100 μ M of GSH ester.
GSH ester only (100 μ M)	5.7	GSH ester only treatment can slightly increase intracellular GSH level by 0.7 mM with the addition of 100 μ M of GSH ester.